# Using Maize $\delta^{15}$N values to assess soil fertility in fifteenth- and sixteenth-century ad Iroquoian agricultural fields

**John P. Hart** *, **Robert S. Feranec**

Research and Collections Division, New York State Museum, Albany, New York, United States of America

* john.hart@nysed.gov

**Data Availability Statement:** All relevant data are within the paper and its Supporting Information files.

**Funding:** The authors received no specific funding for this work.

## Abstract

Native Americans developed agronomic practices throughout the Western Hemisphere adapted to regional climate, edaphic conditions, and the extent of dependence on agriculture for subsistence. These included the mounding or "corn hill" system in northeastern North America. Iroquoian language speakers of present-day New York, USA, and Ontario and Québec, Canada were among those who used this system. While well-known, there has been little archaeological documentation of the system. As a result, there is scant archaeological evidence on how Iroquoian farmers maintained soil fertility in their often-extensive agricultural fields. Using $\delta^{15}$N values obtained on fifteenth- and sixteenth-century AD maize kernels from archaeological sites in New York and Ontario, adjusted to take into account changes that result from charring as determined through experiments, we demonstrate that Iroquoian farmers were successful at maintaining nitrogen in their agricultural fields. These results add to our archaeological knowledge of Iroquoian agronomic practices. Our results also indicate the potential value of obtaining $\delta^{15}$N values on archaeological maize in the investigation of Native American agronomic practices.

## Introduction

Native American farmers developed agronomic practices throughout the Western Hemisphere adapted to climatic and edaphic conditions and the degree of reliance on agricultural production for subsistence. Well known systems of groups who relied heavily on agricultural production include the terraced fields in the Andes of South America [1], the milpa systems of Central America [2], the irrigation systems of the American Southwest [3], and the ridge systems of the upper Mississippi drainage [4]. How these systems functioned is evinced by archaeological investigations of extant features, ethnohistorical documentation, and in some cases, ethnographic documentation. Well known, but less-well understood, are the mounding ("corn hill") systems in temperate northeastern North America. While portions of several fields have been documented (e.g., [5]), and they are recorded in the seventeenth century ethnohistorical record (e.g., [6]), few have been subject to archaeological investigations [7,8]. As a result, there is little direct archaeological information on how these agronomic systems were managed.

**Competing interests:** The authors have declared that no competing interests exist.

Groups using the mounding system in northeastern North American included Iroquoian language family speaking peoples in present-day New York, USA, and Ontario and Québec, Canada. From the fourteenth century AD onwards, Iroquoians lived in villages and towns that were occupied for 20 to 40 years or more [9] and housed hundreds to well over 1,000 individuals [10–12]. A major source of calories for these communities derived from agricultural produce, primarily maize (*Zea mays* ssp. *mays*) [13], but also other crops, including common bean (*Phaseolus vulgaris*), squash (*Cucurbita pepo*), and sunflower (*Helianthus annuus*) [14]. Non-cultivated foods included a wide range of terrestrial animals and plants [14,15], but with freshwater fish being an important source of animal protein for at least some populations [16, 17]. The only domesticated animal present was dog (*Canis lupus familiaris*), which was consumed occasionally at feasts and ceremonies [18].

The typical seventeenth-century AD ethnohistorically documented Iroquoian agricultural field consisted of many small mounds measuring approximately 46–120 cm in diameter and spaced 76–180 cm apart, which were formed with wooden, bone, antler, or stone hoes ([18], p. 178). Each mound contained 3–4 maize plants [18]. Common bean vines often were grown in the same hills with maize whose stalks acted as climbing poles for the bean vines, while squash vines were planted at intervals and occupied spaces around the mounds [19,20]. These fields were highly productive with some estimates suggesting they surpassed that of contemporaneous European and Euro-American farms [21]. While the agronomic [22–24] and nutritional [25] benefits of such polycultures are well established, how earlier Iroquoian farmers maintained the productivity of their fields is not (but see [21]).

Archaeological estimates of the number of cultivated acres needed to feed individual Iroquoian settlement populations are in the hundreds to thousands of acres (e.g., [26], pp. 99–100). Extensive agricultural fields are attested by the seventeenth-century AD ethnohistorical record (e.g., [18]). Analyses of Iroquoian archaeological site locations demonstrate that settlements were sited in locations favorable to agricultural production. For example, Iroquoians in some areas located settlements and fields at elevations that took advantage of thermal belts, which extended growing seasons up to 30 days [27,28]. These analyses also showed that locations selected by some Iroquoian farmers correlated with heavy, moisture-retaining, upland soils with high lime content, which may have facilitated rhizobia bacteria symbionts that provide nitrogen to common bean-plants [28]. Other analyses have found that Iroquoian archaeological settlement sites in New York are located near loamy, well-drained soils [29,30]. Contrary to suggestions that well drained sandy soils were selected by seventeenth-century AD Wendat (Huron) farmers for agricultural fields south of Georgian Bay in Ontario [18], these were simply the most common soils in the area [31].

It is apparent, then, that at least some Iroquoian farmers selected their settlement locations in part with agricultural productivity in mind [26,32]. While the per-acre productivity of Iroquoian agriculture is debatable [18, 21, 33–36], production needed to be sustained throughout the occupational history of each settlement. Given the large number of acres under cultivation it seems unlikely that Iroquoian farmers practiced frequent shifting cultivation throughout the occupational spans of each settlement [37]. Ethnohistoric accounts indicate continuous cultivation of fields for 10–30 years; while 12 years may have been a more likely maximum for seventeenth-century AD Wendat (Huron) fields, where settlements were associated with sandy soils not well suited to maize production [18]. Seventeenth- and eighteenth-century Haudenosaunee (Iroquois) populations in New York, likely practiced continuous cropping over long periods of time, which was achieved by maintaining soil organic matter in naturally fertile Alfisols and Inceptisols [21,36].

Although many thousands of acres were under cultivation at any given time across the Iroquoian region, which had profound impacts on regional biota [32], only a portion of a single

Iroquoian agricultural field located south of Georgian Bay in Ontario with extant mounds has been subjected to archaeological excavation [7]. The mounds at this site had a mean diameter of 102±11 cm and heights of 20 cm and were spaced an average of 140±23 cm apart. Unlike in a buried field remnant in southern New England [8, 38], no evidence for the use of fish as fertilizer in the form of bone and scales was found in the Ontario mounds. The most notable result of soil analysis of the mounds and interstices is that the mounds contained greater amounts of charcoal and at greater depths than did the interstices [8]. This led to the suggestion that the mounds were created from topsoil and wood ash from the initial clearing of the field [8]. It is also likely that charcoal and ash were incorporated into the mounds as a result of annual burning to rid the fields of unwanted vegetation and previous years' crop detritus [8]. One of the primary limiting factors for maize production is nitrogen [39, 40]. Charcoal (biochar) can be effective at increasing or maintaining plant-available soil nitrogen [41–44]. The incorporation of ash may raise the pH of acidic soils [18].

Given the lack of reporting of Iroquoian agricultural fields that date prior to the adoption of European agronomic practices, it is likely that historical Euro-American and Euro-Canadian farming have obliterated those fields. This evident lack of Iroquoian agricultural field preservation precludes direct archaeological assessments of agronomic practices to maintain soil fertility. The seventeenth-century AD Ontario field cannot be taken as representative of agricultural practices across time and throughout the Iroquoian region [21, 36]. We can only surmise that Iroquoian farmers' agronomic practices included efforts to maintain soil fertility adapted to local edaphic conditions based on indirect evidence. This includes site locations, actualistic experiments, ethnohistoric documentation, and general agronomic knowledge, including the evident need for soil amendments to maintain soil fertility over extended periods of time. It is likely that specific practices to maintain soil fertility varied across the Iroquoian region both spatially and chronologically. Intercropping maize with common bean can result in increased availability of nitrogen to maize [24,45–47]. The annual incorporation of unburned crop detritus into the mounds would have maintained soil organic matter, which in turn, provided needed mineralized nitrogen for maize production [21,24, 34, 45–47]; a critical aspect of agronomic systems in naturally fertile temperate soils [48,49]. To the east of the Iroquoian region, archaeologically excavated mounds in a seventeenth-century AD agricultural field on Cape Cod, Massachusetts evinced intensive use of organic inclusions, including fish [8,37]. However, there has been no archaeological evidence directly from Iroquoian mounds themselves to test this hypothesis in the Iroquoian region.

Soil organic matter tends to be depleted in plowed, continuously cropped fields (e.g., [36, 50–52]), thus necessitating the use of fertilizer, such as animal manure, to maintain productivity. Eastern Hemisphere grains (e.g., wheat, *Triticum* spp.) recovered from prehistoric archaeological sites in Europe often exhibit $\delta^{15}N$ values higher than those of uncultivated plants. This evidently resulted from use of draft-animal manure as fertilizer, which increased plant-available nitrogen in plowed fields. Experiments have documented that manure results in high $\delta^{15}N$ values in grains (e.g., [53–57]); high plant $\delta^{15}N$ values are positively correlated globally with high nitrogen content in soil [58,59]. Ammonium ($NH_4+$) and then nitrate ($NO_3-$) production by soil organisms increase with higher N availability. Soil $^{15}N$ proportions increase from the loss of $^{14}N$ through N mineralization, nitrification, leaching, denitrification, and ammonia volatilization, resulting in higher $\delta^{15}N$ values in plants [58,60].

Farmers in eastern North America did not have draft animals—all cultivation was done by hand. There is no ethnohistorical documentation of the use of manure for fertilization by Native Americans prior to the widespread, often forced, adoption of Euro-American agronomic practices [61]. As a result, the use of $\delta^{15}N$ values to assess Native American agronomic practices has not been pursued because it is generally thought that eastern Native American

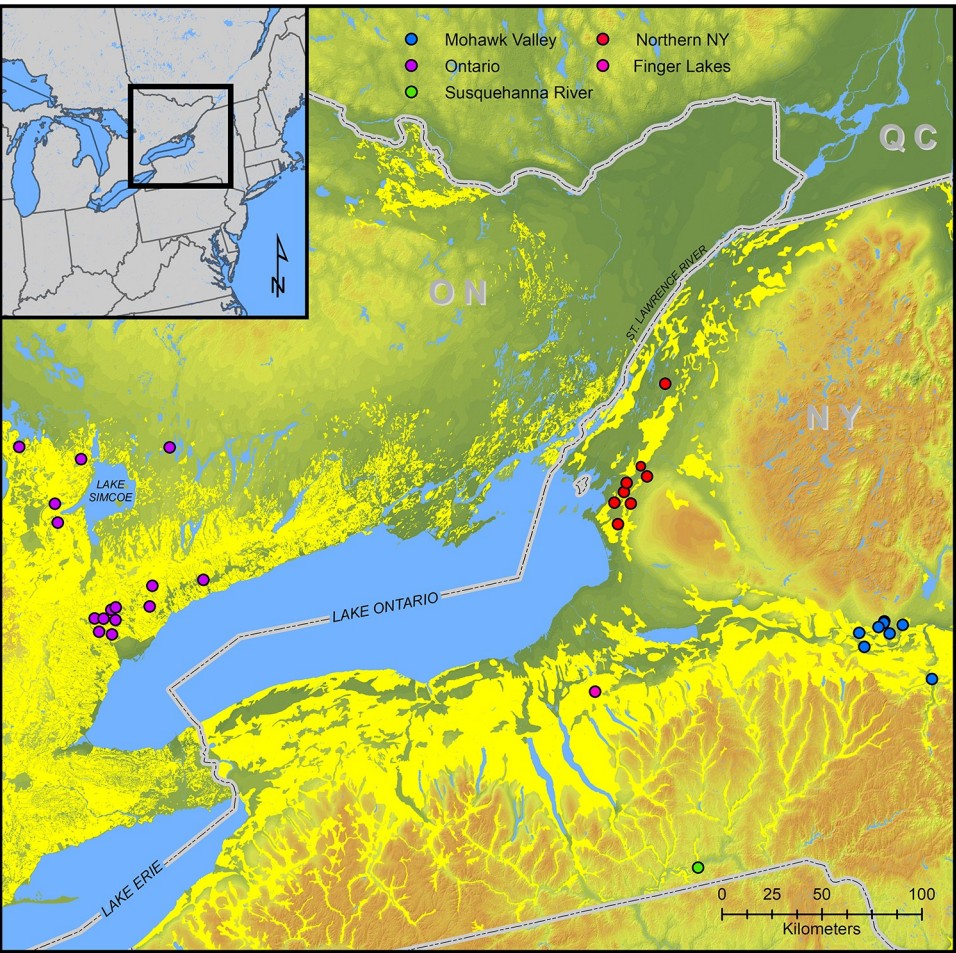

**Fig 1. Locations of Iroquoian archaeological sites from which maize samples originate.** Yellow shading denotes distribution of Alfisols (New York) and Luviols (Ontario). This map was produced in ArcGIS v 10.6 at the New York State Museum in Albany by compiling GIS shape files obtained from publicly available sources including Statistics Canada, the United States Census, the United States Geological Survey, the United States Department of Agriculture, and the Canadian Soil Information Service.

farmers did not practice any form of crop fertilization [18,26]. However, if Iroquoian farmers used varied soil amendments to increase and sustain plant-available nitrogen to maintain fertility of their agricultural fields including unburned crop detritus, then we would expect charred archaeological maize $\delta^{15}N$ values to be consistently higher than values for uncultivated terrestrial plants. Here, we provide the results of $\delta^{15}N$ analyses of charred maize remains, adjusted to take into account the effects of charring, from primarily fifteenth- and sixteenth-century AD Iroquoian sites in portions of New York and Ontario, prior to European ethnohistorical documentation and the consolidation of Wendat (Huron) settlements in an area with sandy soils deficient in natural fertility (Fig 1).

## Results

$\delta^{15}N$ values were obtained on 81 maize kernels and 1 cob fragment from 36 Iroquoian archaeological sites dating primarily to the fifteenth and sixteenth centuries AD in northern New York, the Mohawk River valley of New York, and southern Ontario (Tables 1 and 2, Fig 1).

**Table 1. Archaeological maize samples.**

| Site | Location | Age (century AD) | Samples (n) |
|---|---|---|---|
| Baker | Southern Ontario | 15th | 3 |
| Coulter | Southern Ontario | 16th | 2 |
| Damiani | Southern Ontario | 16th | 2 |
| Dunsmore | Southern Ontario | 15th | 2 |
| Grandview | Southern Ontario | 15th | 2 |
| Hidden Spring | Southern Ontario | 15th | 4 |
| Jarrett-Lahmer | Southern Ontario | 16th | 3 |
| Jones | Southern Ontario | 15th/16th | 1 |
| Mackenzie-Woodbridge | Southern Ontario | 16th | 2 |
| Maynard-McKeown | Southern Ontario | 16th | 2 |
| McNair | Southern Ontario | 15th | 2 |
| New | Southern Ontario | 15th | 1 |
| Parsons | Southern Ontario | 16th | 2 |
| Sopher | Southern Ontario | 16th | 2 |
| Spang | Southern Ontario | 16th | 4 |
| Wapoos | Southern Ontario | 16th | 3 |
| Wellington | Southern Ontario | 14th | 4 |
| Carlos | Northern New York | 15th/16th | 1 |
| Durfee | Northern New York | 15th/16th | 1 |
| Durham | Northern New York | 15th/16th | 1 |
| Morse | Northern New York | 15th/16th | 2 |
| Pine Hills | Northern New York | 15th/16th | 1 |
| Potocki | Northern New York | 15th/16th | 1 |
| Sanford Corner | Northern New York | 15th/16th | 2 |
| Talcott | Northern New York | 15th/16th | 2 |
| Whitford | Northern New York | 15th/16th | 1 |
| Cayadutta | Mohawk Valley | 16th | 4 |
| Garoga | Mohawk Valley | 16th | 3 |
| Getman#1 | Mohawk Valley | 15th | 1 |
| Klock | Mohawk Valley | 16th | 4 |
| Otstungo | Mohawk Valley | 16th | 3 |
| Pethick | Mohawk Valley | 14th | 3 |
| Smith-Pagerie | Mohawk Valley | 15th/16th | 2 |
| Snell | Mohawk Valley | 13th | 3 |
| Roundtop | Susquehanna Valley | 12th–16th | 4 |
| Kelso | Finger Lakes | 14th | 1 |

Values ranged from +0.60 to +9.37‰ with a mean of +5.30±1.54‰ and a median of +5.25‰ (Table 2).

Experiments on Eastern Hemisphere grain kernels, including wheat indicate that charring results in average $\Delta^{15}N$ values of 0.31‰ to 1.00‰ [62,63]. Similar experiments have not been performed on maize. For the present project, samples of contemporary dried Tuscarora White Flour and Dent maize kernels from collections used in previous experiments [64,65] and freeze-dried commercial canned hominy kernels were halved. Following established protocols for experimental charring of maize kernels in non-oxidizing conditions [66,67], one half of each kernel was placed in a loose foil packet, buried in sand within a ceramic crucible, and heated in a muffle furnace at 180˚, 220˚, or 260˚C for 2 h. $\delta^{15}N$ and $\delta^{13}C$ measurements were

**Table 2. $\delta^{15}N$ and $\delta^{13}C$ data for individual maize samples.**

| Lab #[a] | Region | Site | Material | $\delta^{15}N$ | $\delta^{15}N$ adj | $\delta^{13}C$ |
|---|---|---|---|---|---|---|
| UGAMS-37382 | Northern New York | Carlos | kernel | +2.97 | +2.43 | −9.06 |
| UCIAMS-205978 | Northern New York | Durfee | kernel | +5.29 | +4.75 | −9.52 |
| UCIAMS-205971 | Northern New York | Durham | kernel | +4.74 | +4.20 | −9.21 |
| UCIAMS-205977 | Northern New York | Morse | kernel | +6.08 | +5.54 | −8.80 |
| UGAMS-37383 | Northern New York | Morse | kernel | +6.07 | +5.53 | −10.24 |
| UGAMS-37380 | Northern New York | Pine Hills | kernel | +2.67 | +2.13 | −9.17 |
| UCIAMS-205969 | Northern New York | Potocki | kernel | +5.44 | +4.90 | −11.13 |
| UCIAMS-205974 | Northern New York | Sanford Corner | kernel | +9.37 | +8.83 | −9.81 |
| UCIAMS-205975 | Northern New York | Sanford Corner | kernel | +5.46 | +4.92 | −9.21 |
| UGAMS-34445 | Northern New York | Talcott | kernel | +5.70 | +5.16 | −8.80 |
| UGAMS-34446 | Northern New York | Talcott | kernel | +4.87 | +4.33 | −8.80 |
| UGAMS-205972 | Northern New York | Whitford | kernel | +5.86 | +5.32 | −8.87 |
| UCIAMS-205965 | Mohawk Valley | Cayadutta | kernel | +4.85 | +4.31 | −8.56 |
| UCIAMS-205966 | Mohawk Valley | Cayadutta | kernel | +3.98 | +3.44 | −8.88 |
| UCIAMS-205967 | Mohawk Valley | Cayadutta | kernel | +3.06 | +2.52 | −8.88 |
| UCIAMS-205968 | Mohawk Valley | Cayadutta | kernel | +6.48 | +5.94 | −10.24 |
| UCIAMS-218473 | Mohawk Valley | Klock | kernel | +4.29 | +3.75 | −9.35 |
| UCIAMS-218474 | Mohawk Valley | Klock | kernel | +6.76 | +6.22 | −8.78 |
| UCIAMS-218475 | Mohawk Valley | Klock | kernel | +4.55 | +4.01 | −9.50 |
| UCIAMS-218476 | Mohawk Valley | Klock | kernel | +4.72 | +4.18 | −9.05 |
| UCIAMS-218477 | Mohawk Valley | Garoga | kernel | +5.54 | +5.00 | −8.84 |
| UCIAMS-218478 | Mohawk Valley | Garoga | kernel | +4.62 | +4.08 | −9.27 |
| UCIAMS-218479 | Mohawk Valley | Garoga | kernel | +5.80 | +5.26 | −8.55 |
| UCIAMS-218480 | Mohawk Valley | Getman #1 | kernel | +4.33 | +3.79 | −8.07 |
| UCIAMS-218483 | Mohawk Valley | Otstungo | kernel | +5.02 | +4.48 | −9.00 |
| UCIAMS-218487 | Mohawk Valley | Otstungo | kernel | +4.27 | +3.73 | −9.98 |
| UCIAMS-218489 | Mohawk Valley | Otstungo | kernel | +6.71 | +6.17 | −9.11 |
| UCIAMS-218494 | Mohawk Valley | Pethick | kernel | +0.81 | +0.27 | −9.52 |
| UCIAMS-218495 | Mohawk Valley | Pethick | kernel | +2.50 | +1.96 | −8.96 |
| UCIAMS-218496 | Mohawk Valley | Pethick | kernel | +4.73 | +4.19 | −8.96 |
| UCIAMS-218492 | Mohawk Valley | Smith-Pagerie | kernel | +4.66 | +4.12 | −8.68 |
| UCIAMS-218493 | Mohawk Valley | Smith-Pagerie | kernel | +4.75 | +4.21 | −8.93 |
| NYSM-A39855A | Mohawk Valley | Snell | kernel | +7.97 | +7.43 | −9.82 |
| NYSM-A71102 | Mohawk Valley | Snell | kernel | +5.08 | +4.54 | -10.08 |
| NYSM-A71098 | Mohawk Valley | Snell | kernel | +7.08 | +6.54 | −10.46 |
| AA26541/114197 | Susquehanna Valley | Roundtop, 12th/13th cen. | kernel | +0.60 | +0.06 | −8.70 |
| AA21979/114195 | Susquehanna Valley | Roundtop, 14th century | kernel | +2.80 | +2.26 | −8.70 |
| AA26539/114196 | Susquehanna Valley | Roundtop, 15th century | kernel | +2.40 | +1.86 | −8.70 |
| AA21978/114194 | Susquehanna Valley | Roundtop, 16th century | kernel | +4.30 | +3.76 | −8.80 |
| UGAMS-35644 | Finger Lakes | Kelso | kernel | +7.65 | +7.11 | −8.86 |
| UGAMS-32991 | Southern Ontario | Baker | kernel | +6.49 | +5.95 | −9.40 |
| UGAMS-32992 | Southern Ontario | Baker | kernel | +4.41 | +3.87 | −9.31 |
| UGAMS-40364 | Southern Ontario | Barrie | kernel | +5.06 | +4.52 | −9.71 |
| UGAMS-32755 | Southern Ontario | Coulter | kernel | +5.98 | +5.44 | −9.47 |
| UGAMS-32756 | Southern Ontario | Coulter | kernel | +3.84 | +3.30 | −9.31 |
| UGAMS-33005 | Southern Ontario | Damiani | kernel | +5.22 | +4.68 | −9.64 |
| UGAMS-33006 | Southern Ontario | Damiani | kernel | +4.48 | +3.94 | −8.89 |

*(Continued)*

**Table 2.** (Continued)

| Lab #[a] | Region | Site | Material | $\delta^{15}N$ | $\delta^{15}N$ adj | $\delta^{13}C$ |
|---|---|---|---|---|---|---|
| UGAMS-40350 | Southern Ontario | Dunsmore | kernel | +7.66 | +7.12 | −8.23 |
| UGAMS-40351 | Southern Ontario | Dunsmore | kernel | +4.37 | +3.83 | −9.34 |
| UGAMS-40348 | Southern Ontario | Grandview | kernel | +6.25 | +5.71 | −9.24 |
| UGAMS-40347 | Southern Ontario | Grandview | kernel | +3.94 | +3.40 | −8.87 |
| UGAMS-40359 | Southern Ontario | Hidden Spring | kernel | +5.89 | +5.35 | −10.34 |
| UGAMS-40362 | Southern Ontario | Hidden Spring | kernel | +5.83 | +5.29 | −8.79 |
| UGAMS-40361 | Southern Ontario | Hidden Spring | kernel | +5.28 | +4.74 | −9.84 |
| UGAMS-40360 | Southern Ontario | Hidden Spring | kernel | +5.01 | +4.47 | −9.80 |
| UGAMS-40358 | Southern Ontario | Jarrett-Lahmer | kernel | +6.43 | +5.89 | −7.95 |
| UGAMS-40356 | Southern Ontario | Jarrett-Lahmer | kernel | +5.37 | +4.83 | −9.86 |
| UGAMS-40357 | Southern Ontario | Jarrett-Lahmer | kernel | +4.96 | +4.42 | −9.25 |
| UGAMS-40363 | Southern Ontario | Jones | kernel | +7.52 | +6.98 | −9.31 |
| UGAMS-40365 | Southern Ontario | Mackenzie-Woodbridge | kernel | +6.86 | +6.32 | −9.00 |
| UGAMS-40366 | Southern Ontario | Mackenzie-Woodbridge | kernel | +4.20 | +3.66 | −9.66 |
| UGAMS-41528 | Southern Ontario | Maynard-McKeown | kernel | +5.75 | +5.21 | −9.50 |
| UGAMS-41529 | Southern Ontario | Maynard-McKeown | kernel | +6.23 | +5.69 | −9.08 |
| UGAMS-32995 | Southern Ontario | McNair | cob | +4.87 | +4.33 | −9.70 |
| UGAMS-32994 | Southern Ontario | McNair | kernel | +4.61 | +4.07 | −10.32 |
| UGAMS-40353 | Southern Ontario | New | kernel | +6.40 | +5.86 | −9.29 |
| UGAMS-40352 | Southern Ontario | New | kernel | +6.10 | +5.56 | −8.19 |
| UGAMS-33009 | Southern Ontario | Parsons | kernel | +6.54 | +6.00 | −9.81 |
| UGAMS-33008 | Southern Ontario | Parsons | kernel | +4.54 | +4.00 | −9.24 |
| UGAMS-40154 | Southern Ontario | Sopher | kernel | +8.83 | +8.29 | −9.10 |
| UGAMS-40155 | Southern Ontario | Sopher | kernel | +7.58 | +7.04 | −9.28 |
| UGAMS-38398 | Southern Ontario | Spang | kernel | +7.50 | +6.96 | −8.22 |
| UGAMS-37834 | Southern Ontario | Spang | kernel | +7.04 | +6.50 | −8.73 |
| UGAMS-38397 | Southern Ontario | Spang | kernel | +5.94 | +5.40 | −9.44 |
| UGAMS-37833 | Southern Ontario | Spang | kernel | +5.20 | +4.66 | −9.65 |
| UGAMS-41530 | Southern Ontario | Waupoos | kernel | +5.31 | +4.77 | −8.89 |
| UGAMS-41531 | Southern Ontario | Waupoos | kernel | +4.76 | +4.22 | −8.63 |
| UGAMS-41532 | Southern Ontario | Waupoos | kernel | +4.98 | +4.44 | −9.05 |
| UGAMS-40346 | Southern Ontario | Wellington | kernel | +6.98 | +6.44 | −8.99 |
| UGAMS-40343 | Southern Ontario | Wellington | kernel | +5.77 | +5.23 | −9.21 |
| UGAMS-40345 | Southern Ontario | Wellington | kernel | +5.38 | +4.84 | −8.92 |
| UGAMS-40344 | Southern Ontario | Wellington | kernel | +4.31 | +3.77 | −8.55 |

[a] Isotopic measures were obtained on maize samples submitted for previously published AMS radiocarbon dating as indicated in the methods section except those identified by NYSM catalog numbers, which were assayed for this project at the University of Florida Light Stable Isotope Mass Spectrometry Lab. No permits or new permissions were required. AA = University of Arizona AMS Laboratory, UCIAMS = University of California-Irvine Keck Carbon Cycle AMS Laboratory, UGAMS = University of Georgia Center for Applied Isotope Studies.

obtained on fractions of the carbonized and uncarbonized halves. Whole kernels were also carbonized to assess heating effects on kernel integrity [66–68].

Results are presented in Table 3 and S1 Table. Kernels heated for 2 h at 180°C did not fully carbonize and those heated at 260°C for 2 h did not maintain their structural integrity, consistent with outcomes obtained by others [66,67]. As a result, it is unlikely that either would have survived in the archaeological record. The kernels heated at 220°C fully carbonized and

**Table 3. Results of experimental maize charring on $\delta^{15}N$ and $\delta^{13}C$ values.**

| ˚C | Time (h) | n | Mean $\Delta^{15}N$ | Std. Dev. | Median $\Delta^{15}N$ | n | Mean $\Delta^{13}C$ | Std. Dev. | Median $\Delta^{13}C$ |
|---|---|---|---|---|---|---|---|---|---|
| 180 | 2 | 20 | 0.06 | 0.57 | 0.19 | 23 | 0.00 | 0.29 | 0.00 |
| 180 | 24 | 9 | 0.39 | 0.66 | 0.58 | 9 | 0.07 | 0.04 | 0.01 |
| 220 | 2 | 20 | 0.51 | 0.59 | 0.43 | 15 | 0.02 | 0.36 | 0.12 |
| 220 | 24 | 9 | 0.60 | 0.40 | 0.74 | 9 | 0.16 | 0.24 | 0.10 |
| 260 | 2 | 6 | 0.96 | 0.2 | 0.91 | 6 | 0.56 | 0.38 | 0.61 |

maintained their structural integrity appearing much like charred kernels recovered from archaeological sites; the mean $\Delta^{15}N$ for these kernels is 0.51 ‰. The experiments were repeated with different kernels for 24 h at 180˚ and 220˚C, which duplicated the results for kernels heated for 2 hr. The 24 h $\Delta^{15}N$ values are statistically the same as those kernels heated for 2 h (df = 26, t = 0.413, p = 0.6831). Combining the 220˚C 2 and 24 h experiments results in a mean $\Delta^{15}N$ of 0.54±0.53. This value was subtracted from the archaeological maize $\delta^{15}N$ values, and the standard deviation for the archaeological maize mean value was adjusted with the standard deviation of the $\Delta^{15}N$ mean through error propagation calculation. This resulted in a range in $\delta^{15}N$ values for the archaeological kernels of +0.06 to +8.83 ‰, a mean of +4.76±1.63 ‰, and a median of 4.71 ‰.

Terrestrial plants should have $\delta^{15}N$ values 3–4 ‰ lower than terrestrial herbivore bone collagen ([42], p. 3). Archaeologists exploring Neolithic and later agronomic practices in Europe have used this as one means to establish baselines to identify the use of animal manure for crop fertilization. That is, $\delta^{15}N$ values of crop seeds higher than estimated plant browse are interpreted as evidence for manure fertilization [53–55]. Following this line of reasoning we calculated an estimated mean for plant browse from collagen of bone recovered form archaeological sites in the study region. $\delta^{15}N$ values from 227 white tailed deer (*Odocoileus virginianus*) bone collagen samples obtained from Iroquoian archaeological sites in the three areas with sampled maize kernels have a range of +2.8 to +8.6 ‰, a mean of +5.6±1.0 ‰, and a median of +5.5 ‰ (S2 Table). Captive, control-fed, white-tailed deer had a mean $\delta^{15}N$ for antler collagen of +4.29±0.42 ‰ [69]; for a pure $C_3$ diet, the mean was +3.73±0.43‰. Subtracting 25 red deer (*Cervus elaphus*) antler collagen values from same-individual bone collagen values [70] resulted in a mean difference of +0.38±0.37 ‰, suggesting that there is essentially no difference between bone collagen and antler collagen $\delta^{15}N$ values. As a result, we subtracted 4.0 ‰ from individual archaeological deer bone collagen values to estimate average consumed plant $\delta^{15}N$ values, resulting in a mean of +1.6±1.0 ‰ and a median of +1.5 ‰ (Fig 2). This range of results is consistent with similar estimates in Europe based on large herbivore collagen values, ranging from +0.9 to +3.1 ‰ [54].

Subtracting 4.0 ‰ from rabbit/hare (Leporidae) collagen values from southern Ontario Iroquoian archaeological sites (n = 6) resulted in a mean of −0.39±0.73 ‰ and a median of −0.04 ‰. Subtracting the value from woodchuck (*Marmota monax*) collagen values from southern Ontario Iroquoian sites (n = 18) resulted in a mean of −0.93±0.70 ‰ and a median of −0.97 ‰ (S2 Table). These results suggest that the plants consumed by these herbivores had lower $\delta^{15}N$ values than the plants consumed by deer. To be conservative, we used the values for deer to calculate the value for non-cultivated plants in the data evaluation that follows.

Approximately 80% of land plant species are mycorrhizal including maize [71]. Like the majority of these plants, maize is associated with arbuscular mycorrhizae [42,58]. One study found that plants associated with arbuscular mycorrhizae have mean $\delta^{15}N$ values ~2‰ lower than nonmycorrhizal plants, with a mean value of −1.1±0.1 ‰ [58]. Because of the high percentage of arbuscular mycorrhizal plants globally, the estimated browse values should provide

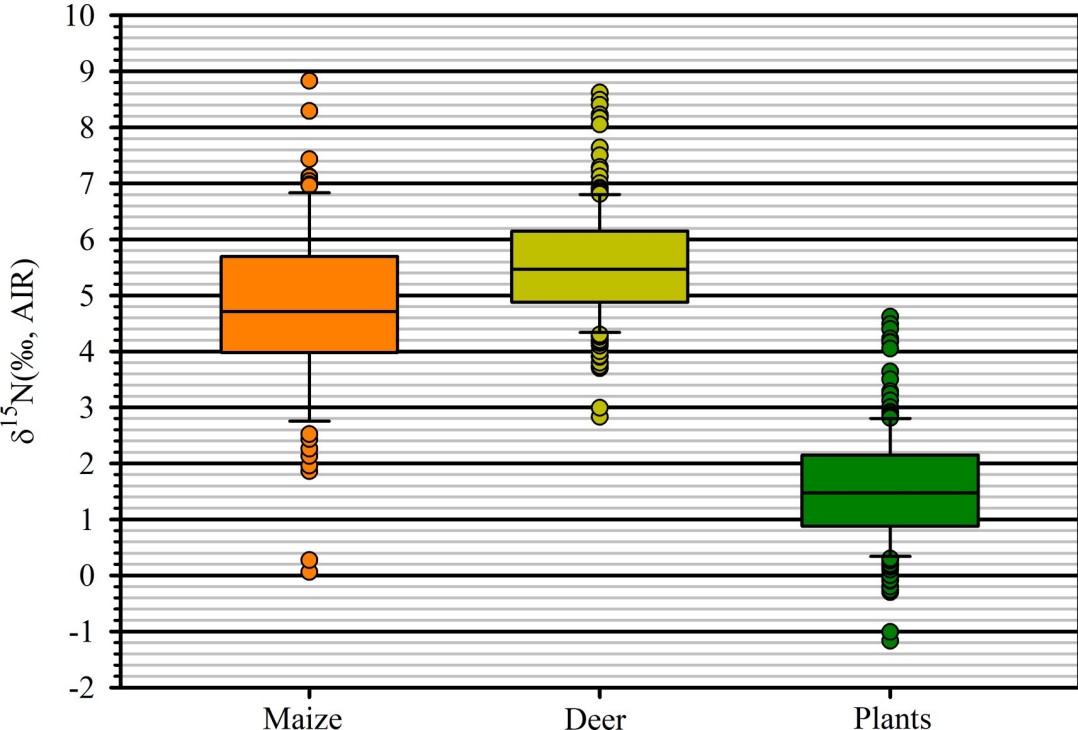

**Fig 2. Box plots of adjusted $\delta^{15}$N values of archaeological maize, archaeological white-tailed deer bone collagen, and estimated deer forage.** The horizontal lines within the boxes are medians, boxes represent the 25th to 75th percentile and whiskers indicate 10th and 90th percentiles.

a reasonable baseline for interpretation of the archaeological maize and are consistent with the lowest $\delta^{15}$N values obtained on archaeological maize kernels. While maize plants may obtain some nitrogen from these fungal symbionts, they primarily obtain mineralized nitrogen from the soil with enhanced phosphorus uptake being the primary benefit to arbuscular mycorrhizal plants [72]. Given the global positive correlation between high soil N and plant $\delta^{15}$N values [58] these values and the overall range in values of +8.77 ‰ suggest that they resulted from varied N pools, with the higher values reflecting high plant-available nitrogen pools. The 72 values greater than two standard deviations above the estimated mean of plant values (>3.6 ‰) is +5.14 ± 1.16 ‰. Two lowest values, +0.06‰ and +0.26‰, are greater than two standard deviations below the mean for estimated plant values, suggesting low soil nitrogen. A t-test indicates that the maize values as a whole are statistically different from the estimated terrestrial plant values based on deer collagen values (p = 0.0000); the maize $\delta^{15}$N values are higher than would be expected for terrestrial plants (difference in means = 3.20‰; Fig 2). Similar results are obtained for the three subregions (Table 4).

The large area encompassed in our study has varied edaphic and climatic conditions, which likely resulted in differing agronomic practices. However, one goal of the varied practices was to maintain productivity to support settlement populations. This included maintaining plant-available nitrogen levels. T-tests of adjusted maize $\delta^{15}$N values between subregions indicate no significant differences between Northern New York and the Mohawk Valley and southern Ontario. There is a significant difference, however, between the Mohawk Valley and southern Ontario; the southern Ontario mean is higher than the Mohawk Valley mean, suggesting the possibility of regional variation (Table 5; Fig 3).

**Table 4. T-test results of maize and estimated plant $\delta^{15}$N values.**

| Region | maize (n) | plant[a] (n) | t | p | Difference in means | Confidence interval (95%) |
|---|---|---|---|---|---|---|
| All | 82 | 227 | 17.445 | 0.0000 | 3.2009 | 2.9002–3.5007 |
| Southern Ontario | 42 | 191 | 19.965 | 0.0000 | 3.4983 | 3.1531–3.8436 |
| Northern New York[b] | 12 | 9 | 4.870 | 0.0000 | 3.0811 | 1.9681–4.1864 |
| Mohawk Valley[b] | 23 | 27 | 10.388 | 0.0000 | 3.6347 | 2.9312–4.3382 |

[a]Estimates (see text)

[b]Exact permutation p, bootstrap confidence interval,.

## Discussion and conclusion

There is little doubt that fifteenth–sixteenth century AD Iroquoian farmers in present-day New York and Ontario needed to maintain the fertility of their extensive, hand-cultivated maize fields over extended periods of time lasting up to several decades. While it is possible that some of the high $\delta^{15}$N values we obtained are the result of initial field clearance involving cutting down and burning trees and other vegetation ([42], p.7), given the probable lengths of time Iroquoian agricultural fields were in continuous cultivation, the effects of these activities on the isotopic compositions of the plants are insufficient to explain the range of $\delta^{15}$N values observed. For example, one study found an initial increase in foliar $\delta^{15}$N values after wildfires, followed by sharp drop offs in the first post-fire decade [73]. A meta-analysis of fire effects on nitrogen pools found that $NH_4^+$ increased immediately following fire and then declined synoptically to pre-fire levels within 3 yr, while $NO_3^-$ increased following the fire peaking at 1 yr and then decreasing to pre-fire levels within 5 yr [74]. An analysis of clearcutting forests also indicated short-lived increase in foliar $\delta^{15}$N values, reaching their peak in 2 yr and falling thereafter [75] (see [42] for an overview). Therefore, clearance involving burning may have resulted in a very short-term pulse of mineralized nitrogen into the soil, possibly elevating plant $\delta^{15}$N values, but this effect would have dissipated well before these fields ceased to be cultivated.

While we do not have direct evidence of soil amendments to increase and maintain plant available nitrogen, Iroquoian agronomic systems were evidently well adapted to local edaphic conditions. All but six (7.3%) of the 82 $\delta^{15}$N maize values used in this analysis exceed the minimal threshold of ~+2.5‰ suggested for identifying fertilization with manure in Europe, and 14 (17.07%) exceed ~+6.0‰ for identifying heavy use of manure [55] (Fig 4). The maintenance of soil organic matter in naturally fertile soils of New York through the incorporation of crop detritus into "corn hills" allowed continuous cropping systems over extended periods of time [21,34]. Long-term incorporation of organic matter with high nitrogen content (>1%), such as common bean and squash vine detritus, can promote the accumulation of nitrogen in soil organic matter [76]. The addition of organic matter to contemporary no-till systems results in high levels of microbial respiration and nitrogen mineralization [77]. Naturally fertile soils containing 4% organic matter can annually produce 90 lbs per acre (102 kg/ha) of plant-available nitrogen [21] in excess of amounts provided under some systems of manure fertilization

**Table 5. T-test results of sub-regional $\delta^{15}$N values.** (Monte Carlo permutation p-values, bootstrap confidence intervals).

| Regions | t | p | Difference in means | Confidence Interval (95%) |
|---|---|---|---|---|
| Northern NY-Ontario | 0.7852 | 0.4372 | 0.3298 | -0.5900–1.3649 |
| Northern NY-Mohawk | 0.8447 | 0.4084 | 0.4797 | -0.6906–1.5610 |
| Mohawk-Ontario | 2.3899 | 0.0204 | 0.8094 | 0.0903–1.5151 |

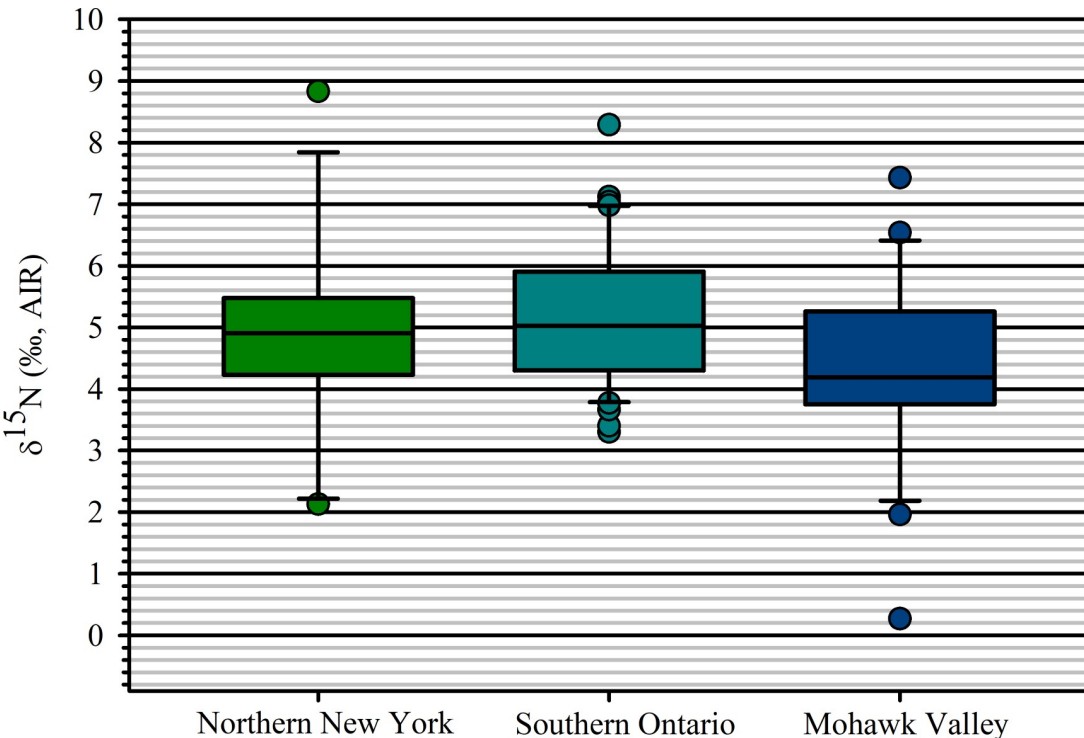

**Fig 3. Box plots of $\delta^{15}$N values of archaeological maize from southern Ontario, northern New York, and the Mohawk Valley.** The solid horizontal lines within the boxes are medians, boxes represent the 25th to 75th percentile and whiskers indicate 10th and 90th percentiles.

[78]. Agronomic systems that included annual burning of fields may have increased plant-available nitrogen [44,79] while the incorporation of charcoal from maize detritus into "corn hills" may have helped maintain plant-available nitrogen [80] and in some cases increased plant nitrogen uptake [81]. Intercropping maize with common bean may also have enhanced the availability of nitrogen to maize [45–47].

Our results suggest that Iroquoian agronomic practices were as effective at providing nitrogen to crops. The contrary interpretations of the seventeenth-century AD Ontario ethnohistorical record [7], which emphasizes the exhaustion of soils after short periods of time, may have been a development of that century when the Wendat (Huron) Confederacy occupied an area with sandy soils having low natural fertility as opposed to areas to the south in Ontario where most settlements were located prior to the seventeenth-century consolidation [31]. Southern Ontario has dominantly Luviols, the equivalent of Alfisols in the U.S.D.A. system [82], the latter of which were exploited by Iroquoian farmers in New York [21,32]. In central New York, Iroquoian village sites, for example, are associated with high fertility Alfisols and areas that experienced high frequencies of evidently anthropogenic fire, presumably from agronomic practices [32,83]. At least some fifteen-and sixteenth-century Iroquoian villages in southern Ontario were sited to take advantage of soils with high natural soil fertility [26]. The low fertility of the acidic soils exploited by Wendat (Huron) farmers in the seventeenth century after consolidation of the Confederacy [18] were not typical of soils exploited by Iroquoian farmers in other areas and times. Mineralization of nitrogen from the microbial breakdown of soil organic matter is negatively affected by lower temperatures in the presence of low soil pH, higher sand content, and lower clay content [84,85], all of which characterize the acidic, sandy soils of the seventeenth-century Wendat (Huron) region [18].

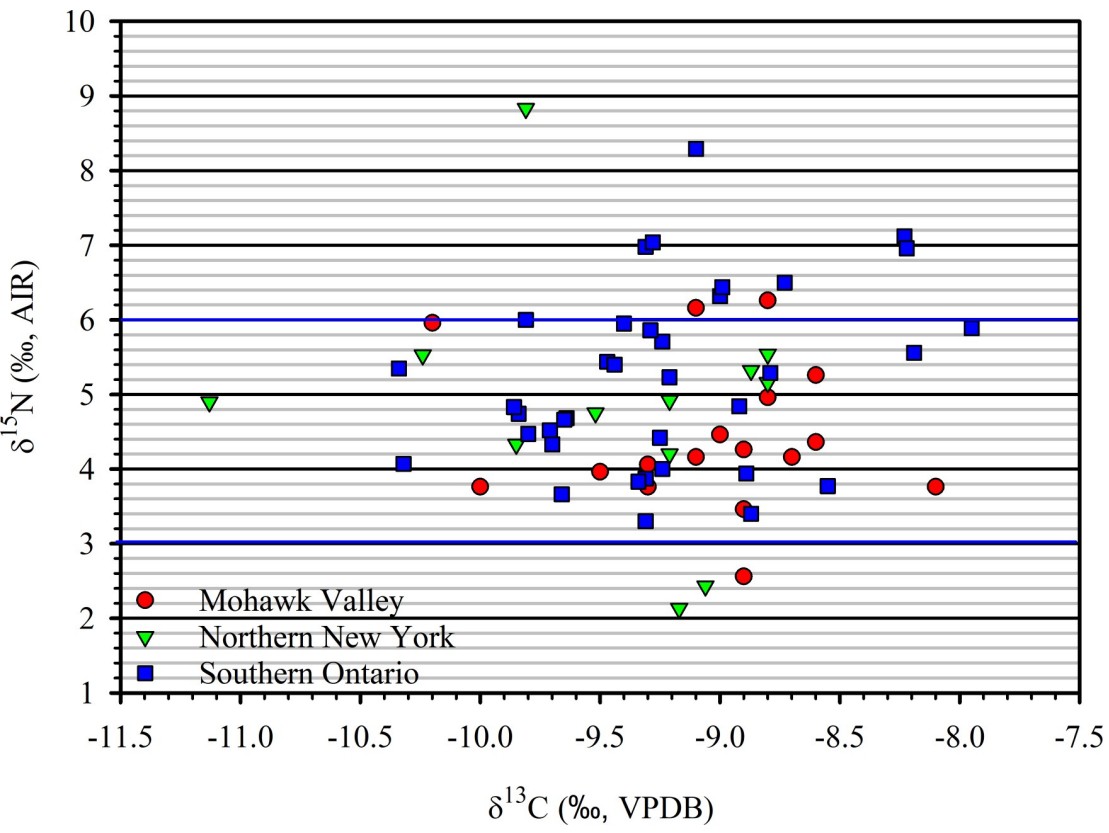

**Fig 4. Scatter diagram of archaeological maize $\delta^{15}$N and $\delta^{13}$C values.** The blue horizontal lines are estimates for boundaries for medium (lower) and high (upper) manuring rates for European Neolithic crops, respectively [53–55].

While Iroquoian farmers may not have used fertilizer in the Eastern Hemisphere sense, our results suggest that as with Native Americans in other regions [86], their agronomic practices maintained plant-available nitrogen, as evinced by high $\delta^{15}$N values. These relatively high $\delta^{15}$N values are suggestive of the addition of soil amendments with $\delta^{15}$N values higher than that of the original soil organic matter. At this early stage it is not possible to say what these soil amendments might have been, but human and/or dog excrement or fish are plausible. Additional studies are required to examine the effects of these practices on plant $\delta^{15}$N values. Our results indicate that reliance on ethnohistoric accounts of seventeenth-century AD agriculture in Ontario to model Iroquoian agriculture in general is unwarranted [21]. The results also demonstrate the utility of obtaining $\delta^{15}$N values on maize as a tool for increasing knowledge of pre-Contact Native American agronomic practices.

## Methods and materials

All statistical analyses were performed in PAST v. 3.25 [87].

Isotopic measures were obtained on maize samples submitted for AMS radiocarbon dating as reported in [88–91] except those obtained on the three samples from the Snell site, which were obtained independently of AMS dating for this project. The values from the Roundtop site were obtained on samples originally reported in [92] on remaining portions of the samples archived at the University of Arizona AMS Laboratory.

The stable isotope results in this study are expressed in standard $\delta$-notation. X = [($R_{sample}$/$R_{standard}$)– 1] * 1000, where X (in units permil, ‰) is $\delta^{13}$C or $\delta^{15}$N and R = $^{13}$C/$^{12}$C or $^{15}$N/$^{14}$N.

The $\delta^{13}$C values are reported relative to the V-PDB standard, while the $\delta^{15}$N values are reported relative to atmospheric $N_2$.

In this study we directly analyzed the $\delta^{13}$C and $\delta^{15}$N values from modern maize kernels. For analysis, dried maize kernels were crushed to a powder using a mortar and pestle then weighed (@ 3.5mg) into tin capsules. The samples were analyzed in the Light Stable Isotope Mass Spectrometry Lab in the Department of Geological Sciences at the University of Florida, Gainesville, FL, USA. Specifically, tin capsules were loaded into a 50-position automated Zero Blank sample carousel on a Carlo Erba NA1500 CNS elemental analyzer. Each sample was combusted at 1020˚C in a quartz column in an oxygen-rich atmosphere. The sample gas was transported and passed through a hot reduction column (650˚C) consisting of elemental copper to remove oxygen in a He carrier stream. The remaining sample gas then passed through a chemical (magnesium perchlorate) trap to remove water followed by a 0.7-meter GC column at 120˚C to separate $N_2$ from $CO_2$. The sample gas next passed into a ConFlo II interface and into the inlet of a Thermo Electron Delta V Advantage isotope ratio mass spectrometer running in continuous flow mode where the sample gas was measured relative to laboratory reference $N_2$ from $CO_2$ gases. Precision for the analyses were $<0.2$‰ for $\delta^{15}$N and $<0.1$‰ for $\delta^{13}$C.

Isotopic analyses of charred archaeological maize were carried out at Keck Carbon Cycle Facility at the University of California Irvine (UCIAMS) or the Center for Applied Isotope Studies at the University of Georgia (UGAMS). Samples at both facilities were subjected to standard acid-base-acid pretreatment. UCIAMS $\delta^{15}$N was measured to a precision of $<0.2$ ‰ using a Fisons NA1500NC elemental analyzer/Finnigan Delta Plus isotope ratio mass spectrometer. At UGAMS, $\delta^{15}$N was measured using an elemental analyzer isotope ratio mass spectrometer to a precision of $<0.1$ ‰.

## Supporting information

**S1 Table. Stable isotope results of maize kernel charring experiments.**
(DOCX)

**S2 Table. Herbivore bone collagen $\delta^{15}$N values used to calculate forage values.**
(DOCX)

## Acknowledgments

UGAMS isotope values provided by Jennifer Birch and Carla S. Hadden were obtained under U.S. National Science Foundation Grant 1727802. UCIAMS isotope values were provided by John Southon. Gregory Hodgins arranged for the Roundtop site AA $\delta^{15}$N measures. Timothy Abel provided the northern New York samples. Access to the Mohawk Valley samples in the New York State Museum's collections was facilitated by Andrea Lain and Jonathan Lothrop. We thank Jane Mt Pleasant for her comments on an earlier draft of this article and Paul Szpak for comments and suggestions on the revision. Susan Winchell-Sweeney provided Fig 1.

## Author Contributions

**Conceptualization:** John P. Hart, Robert S. Feranec.

**Data curation:** John P. Hart.

**Formal analysis:** John P. Hart, Robert S. Feranec.

**Investigation:** John P. Hart, Robert S. Feranec.

**Methodology:** John P. Hart, Robert S. Feranec.

**Validation:** John P. Hart, Robert S. Feranec.

**Writing – original draft:** John P. Hart, Robert S. Feranec.

**Writing – review & editing:** John P. Hart, Robert S. Feranec.

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
