## [Decision Letter · Decision Letter 0]

26 Nov 2019

PONE-D-19-26845

Using Maize δ15N Values to Assess Soil Fertility in Fifteenth- and Sixteenth-Century AD Iroquoian Agricultural Fields

PLOS ONE

Dear Dr. Hart,

Thank you for submitting your manuscript to PLOS ONE. After careful consideration, we feel that it has merit but does not fully meet PLOS ONE’s publication criteria as it currently stands. Therefore, we invite you to submit a revised version of the manuscript that addresses the points raised during the review process.

One of the reviewers has indicated that the interpretation of N isotopic signatures should be improved before publication. Please be sure to address this aspect in the revised manuscript.

We would appreciate receiving your revised manuscript by Jan 10 2020 11:59PM. To enhance the reproducibility of your results, we recommend that if applicable you deposit your laboratory protocols in protocols.io, where a protocol can be assigned its own identifier (DOI) such that it can be cited independently in the future. For instructions see: http://journals.plos.org/plosone/s/submission-guidelines#loc-laboratory-protocols

We look forward to receiving your revised manuscript.

Kind regards,

Remigio Paradelo Núñez

Academic Editor

PLOS ONE

Journal Requirements:

1. In your manuscript, please provide additional information regarding the specimens used in your study. Ensure that you have reported specimen numbers and complete repository information, including museum name and geographic location.

For more information on PLOS ONE's requirements for paleontology and archaeology research, see https://journals.plos.org/plosone/s/submission-guidelines#loc-paleontology-and-archaeology-research.

2. In order to improve statistical reporting, please avoid referring to p-values equal to zero if what is meant is that they are very low.

Reviewers' comments:

Reviewer's Responses to Questions

**Comments to the Author**

1. Is the manuscript technically sound, and do the data support the conclusions?

Reviewer #1: Yes

Reviewer #2: Yes

2. Has the statistical analysis been performed appropriately and rigorously? 

Reviewer #1: Yes

Reviewer #2: Yes

3. Have the authors made all data underlying the findings in their manuscript fully available?

Reviewer #1: Yes

Reviewer #2: Yes

4. Is the manuscript presented in an intelligible fashion and written in standard English?

Reviewer #1: Yes

Reviewer #2: Yes

5. Review Comments to the Author

Reviewer #1: Dear Authors

I appreciate your effort do use d15N values to assess the ancient agriculture practices. Many are the examples of ancient agriculture that was able to support dense populations (Asia) using a variety of soil amendments (in the obvious absence of fertilizers). However, authors might use a more precise interpretation of d15N values to draw their conclusions. Nitrogen isotope natural abundance (expressed here as d15N values) is not an indicator to “plant- available nitrogen levels”. Thus, the assumption of authors that the higher the d15N values (plant), the higher the plant-available nitrogen levels” is not correct. I fact, the N transformation processes lead to fractionation of nitrogen isotope and the N loss (in soli or manure storage) results in an increasing of d15N values. Manures have high d15N values manly due to N loss as ammonia (volatilization). Plants tend to mirror the d15N values of available N in soil which can reflect the N source, i.e. manure and plant residues and biological N fixing most of the time (I am excluding NH3 deposition for this case). Thus, high d15N indicates loss of N from the system (i.e. soil). However, soil amended with animal manure tends to show high d15N values. I recommend to the authors to bring to support their discussion two good reviews: CHOI, W. J. et al. Synthetic fertilizer and livestock manure differently affect δ15N in the agricultural landscape: A review. Agriculture, Ecosystems & Environment, v. 237, p. 1-15, 2017, and Martinelli LA, Piccolo MC, Townsend AR, Vitousek PM, Cuevas E, McDowell W et al., Nitrogen stable isotopic composition of leaves and soil: tropical versus temperate forests. Biogeochemistry 46:45–65 (1999). The last shows d15N values of soil and leaves (temperate vs. tropical forests). Tropical forests have higher d15N values than Temperate Forests.

Despite of my opinion that the authors might refine the discussion using the right interpretation of d15N values as above mentioned, they are right suggesting a more relevant role of N-fixing from common bean instead use of animal manures based on d15N values obtained. But at this point the paper also needs to give more information about the d15N of N-fixing plants (ranges, common values, etc). This is crucial to support their right conclusions. Thus, I recommend the publication of this manuscript after revision.

Best regards

Reviewer

Reviewer #2: The following text is the same as the attached file for the Editor and Authors.

Review of PONE-D-19-26845

Title: Using Maize δ15N Values to Assess Soil Fertility in Fifteenth- and Sixteenth-Century AD Iroquoian Agricultural Fields

Authors: John P. Hart and Robert S. Ferenec

General Comments to Editor and Authors:

This paper provides a compelling case for Iroquoian farmers in New York State, USA and Ontario and Québec in Canada having maintained the fertility of their agricultural fields during the 15th and 16th centuries AD. Specifically, the authors obtained δ15N and δ13C values on maize kernels from >60 Iroquoian archaeological sites, and performed statistical analysis on these isotopic data, to elucidate upon the fertility of the soils that once supported maize at these sites. The authors analyzed these isotopes in the absence of preserved Iroquoian fields, therefore, δ15N and δ13C values on maize kernels can be viewed as an indirect proxy for soil productivity (an idea not fully fleshed out in the manuscript, see below). To strengthen the hypothesis that Iroquoian farmers burned their maize fields after harvest to input plant-available nitrogen, suggested from analysis of the archaeological material, the authors conducted an experimental study. The authors charred and performed isotopic analysis upon modern maize kernels as well as grass seeds collected in the 1930s-1940s, which validated their interpretations of the isotopic data based on Iroquoian archaeological material. The authors’ results were also compared to the experimental data on wheat kernels derived from archaeological sites in the Eastern Hemisphere.

The key finding of this research, presented for the first time in this paper (to my knowledge), is that the amount of nitrogen input into the soils through burning by Iroquoian farmers during the 15th and 16th centuries AD was comparable to adding manure to agricultural fields by Neolithic farmers in Europe. This finding is highly significant and requires to be more strongly highlighted in the paper (below I indicate where in the manuscript). Additionally, I recommend that the authors expound upon other aspects of their findings and the broader significance of their research in their revision of this manuscript.

In my opinion, this manuscript is worthy of publication in PLOS ONE pending minor revisions. The paper describes original research not published elsewhere. The theoretical and methodological approaches and explanations are robust and the statistical analyses are excellent; for example, the reasoning expressed in the paragraph on P. 7, Lns 188-205, is exceptional. The conclusions are well supported by the data. The research adheres to integrity standards, and the data will be available upon publication. The paper is also well written. Below are more specific comments to aid the authors when revising this paper.

Specific Comments (my suggested textual additions are in bolded red font):

Page # Line # Comments:

* My textual additions are in red font.

1 6, 32 Abstract & Introduction: insert “State, USA” after New York, and “Canada” after “Québec” [use the letter e with an acute accent, which is the official spelling of this province’s name], to read “New York State, USA and Ontario and Québec, Canada,”.

1 9-12 The authors did not mention an experimental component of their research, where they charred modern maize kernels and 20th century grass seeds to valid their isotopic values obtained from maize kernels obtained from archaeological results. Therefore, a short sentence regarding this work should be added to the Abstract.

1 13 Abstract: add a phrase to the end of this sentence, or an additional sentence, that identifies the broader significance of this research beyond Iroquoian studies. Here is a suggestion: “These results…Iroquoian agronomic practices, and identified that Iroquoian farmers augmented nitrogen through burning, producing values comparable to European Neolithic farmers who utilized manure to upkeep soil productivity.”

1 46 Provide the source for this sentence about 3-4 maize plants per mound.

3 98-99 Write a sentence or two that explains why there is an “apparent current lack of Iroquoian agricultural field preservation.” I suggest that the authors move up the first sentence of the following paragraph (Ln 116-117) about depletion of soil organic matter to Ln 98. Also, add a statement that the areas where Iroquoian agriculture fields were once located have been heavily farmed by Euro-Americans and Euro-Canadians in subsequent decades, thereby obliterating the soil characteristics of the fields farmed centuries ago by indigenous populations.

3 101-105 Please divide this long sentence into two.

3 113 Insert “directly from the mounds themselves” between “evidence” and “to test,” given that the paper does provide archaeological evidence to test this hypothesis.

3 116 Insert an introductory sentence, before the one beginning with “Eastern Hemisphere grains…”, which explains that in the absence of preserved Iroquoian agriculture fields one can use δ15N values obtained from cultigens as an indirect proxy for past soil productivity.

6 152 Table 2: at the bottom of this table, provide footnotes with details, including the full names, pertaining to the two labs used, UGAMS and UCIAMS (which are identified on p. 12). Note: I approve of the authors using only two labs to obtain these δ15N and δ13C data, to minimize any possible data discrepancy between them. While looking at the data in this table, I did not see any indication of variation in the results produced between the two labs, as both labs gave a similar range of values for each isotope. Additionally, these AMS labs have reputations for producing outstanding (reliable) results.

6 155 The colored dots are not identified in Fig. 1. Therefore, the authors should insert a legend explaining the color scheme for these dots, which indicate site locations in southern Ontario, northern New York, and the Mohawk Valley. This information could be added to the figure caption, but a better option is to embed a legend in this figure.

6 161 Identify the grain as being wheat (Triticum spp.).

6 173-175 Move this sentence starting with “The kernels heated at 220°C fully carbonized…” to integrate with the sentence starting on Ln 177, “Those heated to 220°C…” to reduce redundancy.

7, 9 201, 250 Is “cal” necessary, given that BC/AD are used?

7 211 Mention the three subregions at the end of this caption for Fig. 2 (d).

8 213 Replace Quebec with Québec.

10 285 Delete “in” before “can.”

11 315 Insert “involving fire” after agronomic practices.

6. PLOS authors have the option to publish the peer review history of their article (what does this mean?). If published, this will include your full peer review and any attached files.

Reviewer #1: Yes: Caio T Inácio

Reviewer #2: No

---

## [Author Response · Author response to Decision Letter 0]

13 Feb 2020

Responses to Reviewers

Reviewer 1

I appreciate your effort do use d15N values to assess the ancient agriculture practices. Many

are the examples of ancient agriculture that was able to support dense populations (Asia) using

a variety of soil amendments (in the obvious absence of fertilizers). However, authors might

use a more precise interpretation of d15N values to draw their conclusions. Nitrogen isotope

natural abundance (expressed here as d15N values) is not an indicator to “plant- available

nitrogen levels”. Thus, the assumption of authors that the higher the d15N values (plant), the

higher the plant-available nitrogen levels” is not correct. I fact, the N transformation processes

lead to fractionation of nitrogen isotope and the N loss (in soli or manure storage) results in an

increasing of d15N values. Manures have high d15N values manly due to N loss as ammonia

(volatilization). Plants tend to mirror the d15N values of available N in soil which can reflect

the N source, i.e. manure and plant residues and biological N fixing most of the time (I am

excluding NH3 deposition for this case). Thus, high d15N indicates loss of N from the system

(i.e. soil). However, soil amended with animal manure tends to show high d15N values. I

recommend to the authors to bring to support their discussion two good reviews: CHOI, W. J.

et al. Synthetic fertilizer and livestock manure differently affect δ15N in the agricultural

landscape: A review. Agriculture, Ecosystems & Environment, v. 237, p. 1-15, 2017, and

Martinelli LA, Piccolo MC, Townsend AR, Vitousek PM, Cuevas E, McDowell W et al.,

Nitrogen stable isotopic composition of leaves and soil: tropical versus temperate forests.

Biogeochemistry 46:45–65 (1999). The last shows d15N values of soil and leaves (temperate

vs. tropical forests). Tropical forests have higher d15N values than Temperate Forests.

Despite of my opinion that the authors might refine the discussion using the right

interpretation of d15N values as above mentioned, they are right suggesting a more relevant

role of N-fixing from common bean instead use of animal manures based on d15N values

obtained. But at this point the paper also needs to give more information about the d15N of N fixing plants (ranges, common values, etc). This is crucial to support their right conclusions.

Thus, I recommend the publication of this manuscript after revision.

Thank you for your comments. We have addressed the issue of nitrogen loss and δ15N in lines 125 through 129 and the last paragraph. We appreciate your suggestions for additional references. While we did not add these specific we have added citations to several additional sources throughout the document that we feel are more pertinent to the issue at hand and revised/added text that draws on information contained within those surces.

Reviewer 2

1 6, 32 Abstract & Introduction: insert “State, USA” after New York, and “Canada” after

“Québec” [use the letter e with an acute accent, which is the official spelling of this province’s

name], to read “New York State, USA and Ontario and Québec, Canada,”.

We have made these modifications except adding State after New York.

1 9-12 The authors did not mention an experimental component of their research, where they

charred modern maize kernels and 20th century grass seeds to valid their isotopic values

obtained from maize kernels obtained from archaeological results. Therefore, a short sentence

regarding this work should be added to the Abstract.

Done

1 13 Abstract: add a phrase to the end of this sentence, or an additional sentence, that

identifies the broader significance of this research beyond Iroquoian studies. Here is a

suggestion: “These results…Iroquoian agronomic practices, and identified that Iroquoian

farmers augmented nitrogen through burning, producing values comparable to European

Neolithic farmers who utilized manure to upkeep soil productivity.”

Done

1 46 Provide the source for this sentence about 3-4 maize plants per mound.

Done

3 98-99 Write a sentence or two that explains why there is an “apparent current lack of

Iroquoian agricultural field preservation.” I suggest that the authors move up the first sentence

of the following paragraph (Ln 116-117) about depletion of soil organic matter to Ln 98. Also,

add a statement that the areas where Iroquoian agriculture fields were once located have been

heavily farmed by Euro-Americans and Euro-Canadians in subsequent decades, thereby

obliterating the soil characteristics of the fields farmed centuries ago by indigenous

populations.

Done

3 101-105 Please divide this long sentence into two.

Done 

3 113 Insert “directly from the mounds themselves” between “evidence” and “to test,” given

that the paper does provide archaeological evidence to test this hypothesis.

Done

3 116 Insert an introductory sentence, before the one beginning with “Eastern Hemisphere

grains…”, which explains that in the absence of preserved Iroquoian agriculture fields one can

use δ15N values obtained from cultigens as an indirect proxy for past soil productivity.

We have chosen to keep the beginning of this paragraph as in the original.

6 152 Table 2: at the bottom of this table, provide footnotes with details, including the full

names, pertaining to the two labs used, UGAMS and UCIAMS (which are identified on p. 12).

Done

6 155 The colored dots are not identified in Fig. 1. Therefore, the authors should insert a

legend explaining the color scheme for these dots, which indicate site locations in southern

Ontario, northern New York, and the Mohawk Valley. This information could be added to the

figure caption, but a better option is to embed a legend in this figure.

There is now a key to the colors in the figure.

6 161 Identify the grain as being wheat (Triticum spp.).

Done 

6 173-175 Move this sentence starting with “The kernels heated at 220°C fully carbonized…”

to integrate with the sentence starting on Ln 177, “Those heated to 220°C…” to reduce

redundancy.

We have rewritten this paragraph to clarify and avoid redundancy.

7, 9 201, 250 Is “cal” necessary, given that BC/AD are used?

We have deleted the “cal”.

7 211 Mention the three subregions at the end of this caption for Fig. 2 (d).

We have added a key to the dot colors.

8 213 Replace Quebec with Québec.

Done

10 285 Delete “in” before “can.”

We eliminated this table after receiving additional low values on archeological maize, which serve as a better baseline for understanding the elevated values on archaeological maize.

11 315 Insert “involving fire” after agronomic practices.

Because this is only one possible aspect of the agronomic practices be have declided to modify this sentence.

---

## [Editor Report · Decision Letter 1]

13 Mar 2020

Using Maize δ15N Values to Assess Soil Fertility in Fifteenth- and Sixteenth-Century AD Iroquoian Agricultural Fields

PONE-D-19-26845R1

Dear Dr. Hart,

We are pleased to inform you that your manuscript has been judged scientifically suitable for publication and will be formally accepted for publication once it complies with all outstanding technical requirements.

With kind regards,

Remigio Paradelo Núñez

Academic Editor

PLOS ONE

Additional Editor Comments (optional):

Dr. Hart,

Thank you for having considered the questions raised by the reviewers during the revision of your manuscript. Your paper is now acceptable for publication. Congratulations.
---

## [Editor Report · Acceptance letter]

18 Mar 2020

PONE-D-19-26845R1 

Using Maize δ15N Values to Assess Soil Fertility in Fifteenth- and Sixteenth-Century AD Iroquoian Agricultural Fields 

Dear Dr. Hart:

I am pleased to inform you that your manuscript has been deemed suitable for publication in PLOS ONE. Congratulations! Your manuscript is now with our production department. 

With kind regards,

on behalf of

Dr. Remigio Paradelo Núñez 

Academic Editor

PLOS ONE